# Study on the Mechanism of Phenylacetaldehyde Formation in a Chinese Water Chestnut-Based Medium during the Steaming Process

**DOI:** 10.3390/foods12030498

**Published:** 2023-01-21

**Authors:** Yanan Lin, Guanli Li, Shujie Wu, Xiaochun Li, Xiujuan Luo, Dexin Tan, Yanghe Luo

**Affiliations:** 1College of Biological and Chemical Engineering, Guangxi University of Science and Technology, Liuzhou 545006, China; 2Guangxi Key Laboratory of Health Care Food Science and Technology, Hezhou University, Hezhou 542899, China

**Keywords:** Chinese water chestnut, phenylacetaldehyde, Maillard reaction, SPME-GC-MS

## Abstract

The white pulp of the Chinese water chestnut (CWC) is crisp and sweet with delicious flavours and is an important ingredient in many Chinese dishes. Phenylacetaldehyde is a characteristic flavoured substance produced in the steaming and cooking process of CWC. The steaming process and conditions were simulated to construct three Maillard reaction systems which consisted of glucose and phenylalanine, and of both alone. The simulation results showed that glucose and phenylalanine were the reaction substrates for the formation of phenylacetaldehyde. The intermediate α-dicarbonyl compounds (α-DCs) and the final products of the simulated system were detected by solid-phase microextraction (SPME) and gas chromatography–mass spectrometry (GC-MS) methods. Through the above methods the formation mechanism of phenylacetaldehyde is clarified; under the conditions of the steaming process, glucose is caramelized to produce Methylglyoxal (MGO), 2,3-Butanedione (BD), Glyoxal (GO) and other α-DCs. α-DCs and phenylalanine undergo a Strecker degradation reaction to generate phenylacetaldehyde. The optimal ratio of the amount of substance of glucose to phenylalanine for Maillard reaction is 1:4. The results can provide scientific reference for the regulation of flavour substances and the evaluation of flavour quality in the steaming process of fruits and vegetables.

## 1. Introduction

The Chinese water chestnut (CWC) is a characteristic agricultural product in China, especially in Guangxi. The annual output of CWC from China exceeds one million tons, and the commodity trade accounts for more than 99% of the world. This is mainly produced in Hezhou and Guilin of Guangxi. CWC is very nutritious yet low in calories. A 100 g raw fresh CWC sample provides about 97 kcal energy. It consists of 73% water, 24% carbohydrate, 5% sugars, 3% dietary fiber, 1.4% protein, 0.1% lipid and a small amount of minerals and vitamins [1]. The white pulp is crisp and sweet with delicious flavours both raw and cooked, and is an important ingredient of many Chinese dishes. Li et al. [2] found through the research of electronic nose, SPME-GC-MS and ultra-performance liquid chromatography-tandem mass spectrometry (UPLC-MS/MS) technology that the characteristic flavour substances produced by CWC in the steaming and cooking process are mainly phenylacetaldehyde, nonanal, decanal, trans-2-octenal, etc. Among them, phenylacetaldehyde has a strong scent of hosta flower [3], which is the main source of the sweet smell of CWC after steaming and cooking. The formation mechanism of flavour substances refers to the processes of cracking, polymerization, condensation, oxidation, dehydration and other reactions between substrates, and flavour substances are finally formed through a series of molecular transformations [4,5]. The methods to study the formation mechanism of flavour substances mainly include simulation experiments in vitro and isotope tracing. A simulation experiment in vitro was used to infer the molecular path of the formation of flavour substances by detecting the substances produced by different simulation systems that were in contrast with each other. Zhao et al. [6] deduced the formation mechanism of thiophene, pyridine, mercaptan and furan compounds of an alkyl chain in an oxidized chicken fat–glutathione–glucose system by comparing the oxidized chicken fat–glutathione–glucose reaction system with a glutathione–glucose reaction system and the compounds produced by the oxidative degradation of chicken fat. The isotopic tracing method involves using radioactive or stable tracer atoms or compounds to study the transfer and transformation laws of tracked compounds [7], so as to infer the formation mechanism of flavour substances. Liu [8] studied the carbon skeleton source of pyrazine generated in a Maillard reaction system by using the ^13^C labelled xylose isotope tracing method. It is of great practical significance to study the formation mechanism of food flavour substances for regulating the formation of flavour substances and evaluating flavour quality. At present, there is no report on the formation mechanism of CWC flavour substances.

The Maillard reaction is a series of oxidation, cyclization, dehydration and polymerization reactions between carbonyl compounds and amino compounds and is an important reaction that forms food flavour substances [9]. Dry jujube contains aldehydes, hydrocarbons, acids, heterocycles and other flavour substances; among them, 2-pentylfuran, furfural and 2-methylpyrazine are derived from the Maillard reaction [10]. Tang et al. [11] conducted the Maillard reaction of flammulina velutipes enzymatic hydrolysate with glucose and detected 69 flavour substances, including 2-methylbutyraldehyde, 3-methylbutyraldehyde and phenylacetaldehyde, through GC-MS. Zhang [12] detected 35 flavour substances in the clear stewed tube bone soup, including benzaldehyde, 2-undecanone and pyridine from the Maillard reaction. Chen et al. [5] detected decanal, acetaldehyde, octanal, nonanal and other aroma components in the Maillard reaction simulation system of seven amino acids with maltose. At present, there is no report on the formation mechanism of flavour compounds in fruit and vegetable steaming processes based on the Maillard reaction. The Strecker degradation of amino acids mainly occurs in the advanced stage of the Maillard reaction, and is an important source of flavour substances. In this process, amino acids react with α-dicarbonyl compounds (α-DCs) to form aldehydes, carbon dioxide and α-amino ketones through oxidative decarboxylation [13], and the aldehydes and acids produced by the Strecker degradation of amino acids have strong aromogenic potential. α-DCs are intermediate products with high reactivity in the Maillard reaction, and are also the main precursor of flavour substances. The generation of α-DCs mainly includes a caramelization reaction [14] and the Maillard reaction [15]. The caramelization reaction is the main pathway for the formation of α-DCs [16,17], and includes carbohydrate degradation and oxidation reactions, such as glucose degradation to 3-deoxyglucone aldehyde (3-DG), and oxidation to glucose ketoaldehyde (G). Thornalley et al. [18] detected 3-DG produced by glucose degradation at 37 °C. In the Maillard reaction pathway, the reaction products of sugars and amino acids form macromolecule α-DCs, such as G [19], 1-DG and 3-DG [9], through Amadori rearrangement, oxidation, deamination and enolization. Hofmann et al. [20] detected 3-DG with 1 mmol phenylalanine and 1 mmol glucose heated at 100 °C for 30 min. When the pH of the Maillard reaction is less than 7, Amadori rearrangement products tend to undergo a 1,2-alcoholization reaction to form 3-DG; when the pH of the reaction is greater than 7, the rearrangement products tend to undergo a 2,3-alkenyl alcoholization reaction to form 1-DG. These long-chain α-DCs are easily broken into small-molecule α-DCs, such as glyoxal (GO), acetone aldehyde (MGO), etc.

The accurate detection of α-DCs and their formation routes is the key to study the formation mechanism of flavour substances in the Maillard reaction. Tian et al. [21] reported that α-DCs such as GO and MGO reacted with L-leucine, respectively, to form isovaleric aldehyde. Wang et al. [22] reported that o-phenylenediamine (OPD) was added to beer as an inhibitor to prevent the reaction between α-DCs and amino acids, but no flavour substance for Strecker aldehyde was detected. Currently, the main methods for the detection of α-DCs include high pressure liquid chromatography-ultraviolet spectrometry (HPLC-UV) [23], HPLC-MS [24], GC-MS [25], etc. HPLC-UV can detect α-DCs introduced into chromophore by derivation, but an extensive derivation time will affect the accuracy of detection. HPLC-MS is an effective method to separate and analyse complex organic mixtures, but short-chain α-DCs cannot be detected, and by-products may be produced during the determination of samples with high sugar content, affecting the accuracy of the detection results [26]. GC-MS has the advantages of high sensitivity, wide detection range, etc. It is used to detect volatile compounds, which can make up for the shortcoming of HPLC-MS in detecting short-chain α-DCs [27].

Compared with fresh CWC, the flavour of CWC becomes more intense after steaming, which may be because the substances contained in CWC crack or react with each other at high temperature to produce flavour substances [2]. The substances involved in the reaction can be preliminarily determined by comparing the content changes before and after steaming. Our team used the detection method in reference [28] to detect the substances contained in fresh CWC and steamed CWC by HPLC-MS/MS. According to the changes in its peak area, the contents of phenylalanine and glucose in steamed CWC were significantly reduced, which may be the possible substrate of the characteristic flavour substance phenylacetaldehyde. Our previous research showed that the peak area of L-phenylalanine decreased from 1,360,000.00 to 1,233,333.33 after the CWC was steamed, the peak area of Phe-Phe decreased from 1,366,666.67 to 1,173,333.33, the peak area of N-Acetyl-L-phenylalanine decreased from 1,836,666.67 to 1,306,666.67, the peak area of D-(+)-Phenylalanine decreased from 3,620,000.00 to 3,346,666.67, the peak area of glucosamine decreased from 537,000.00 to 491,666.67, the peak area of N-Acetyl-D-glucosamine decreased from 711,666.67 to 650,666.67, the peak area of D-Glucose 6-phosphate decreased from 2,136,666.67 to 1,406,666.67, and the peak area of Glucose-1-phosphate decreased from 2,156,666.67 to 1,426,666.67. As can be seen from the detection data, the content decreased greatly. According to reference [29], phenylacetaldehyde is generated from glucose and phenylalanine by the Maillard reaction. In this study, glucose and phenylalanine were selected as possible substrates for the characteristic flavour substance phenylacetaldehyde, and three simulation systems were established to confirm the reaction substrates for phenylacetaldehyde. We simulated the contents of glucose and phenylalanine in CWC and its steaming processing conditions, captured the small molecules formed in the reaction process with OPD, detected α-DCs and the final products with SPME-GC-MS, and determined the formation substrate and molecular transfer pathway of phenylacetaldehyde to clarify the formation mechanism of the flavour substance phenylacetaldehyde. The results could provide a scientific reference for study on the controlled release of flavour substances and the evaluation and regulation of flavour quality in the process of CWC steaming, and could have important significance for improving the flavour quality of processed CWC products.

## 2. Experimental Instruments and Methods

### 2.1. Experimental Instruments and Reagents

Electronic balance: PTX-FA110S, Fuzhou HZ Electronic Technology Co., Ltd. (Fuzhou, China); induction cooker: PEN3, Beijing Yingsheng Hengtai Technology Co., LTD. (Beijing, China); magnetic stirrer: MRHI-TEC (CN), Heidolph (Schwabach, Germany); solid-phase microextraction head: 50/30 μm DVB/CAR/PDMS Gray, Supelco (Bellefonte, PA, USA); GC-MS co-detector: TRACE 1300-ISQQD, Agilent (Santa Clara, CA, USA); headspace bottle: 20 mL, Supelcoe (Bellefonte, PA, USA); glucose: analytical pure, Guangdong Guanghua Technology Co., LTD. (Shantou, China); L-phenylalanine: biotechnology grade, Shanghai Maclin Biochemical Reagents Co., LTD. (Shanghai, China); disodium hydrogen phosphate: analytical pure, Tianjin Zhiyuan Chemical Reagent Co., LTD. (Tianjin, China); sodium dihydrogen phosphate: analytical pure, Tianjin Zhiyuan Chemical Reagent Co., LTD. (Tianjin, China); O-phenylenediamine (OPD): analytical pure, Tianjin Damao Chemical Reagent Factory (Tianjin, China); 2,4,6-trimethylpyridine (TMV): GC&T, Tokyo Chemical Industry Co., LTD. (Tokyo, Japan); methanol: chromatographic grade, Shanghai Maclin Biochemical Reagents Co., LTD. (Shanghai, China).

### 2.2. Construction of Maillard Reaction Simulation System

Referring to the data on the content of glucose and phenylalanine in CWC detected by the research group using HPLC with GB 5009.8-2016 and GB 5009.124-2016, respectively, we built three simulation systems.

System 1: 2.408/100 g glucose and 0.011/100 g phenylalanine were weighed and dissolved in a 0.2 mol/L buffer solution of disodium hydrogen phosphate and sodium dihydrogen phosphate at pH 5.4, and then fixed at a volume of 100 mL.

System 2: 2.408/100 g glucose was weighed and dissolved in the buffer solution above.

System 3: 0.011/100 g phenylalanine was weighed and dissolved in the buffer solution above.

Accurately, 2.408 g of the glucose and 0.011 g of the phenylalanine were weighed and dissolved in 0.2 mol/L at pH 5.4 disodium hydrogen phosphate and sodium dihydrogen phosphate buffer solution, respectively, and the volume was fixed to 100 mL to obtain two Maillard reaction control simulation systems (2 and 3).

Then, 5 mL solution of the above three simulated systems was removed with a pipette gun and placed in three 20 mL headspace bottles; the internal standard TMV was dissolved in methanol to make the final volume concentration of the solution 0.2 μL/mL, and the solution was accurately absorbed and added to the solution in the headspace bottle with 1 μL; an ordinary steamer and an induction cooker were used for steaming. The output power of the induction cooker was 800 W. After the water boiled, the headspace bottle was put into the steamer for steaming and was timed. The steaming times were 0, 10, 20 and 30 min. The steam environment in the steamer at this time was 100 °C. After the steaming, the test sample was obtained.

### 2.3. SPME-GC-MS Detection

The steaming flask was placed in a constant temperature water bath at 80 °C and stirred with 100 rpm magnetic force. The 50/30 μm DVB/CAR/PDMS Gray extraction head, aged at 280 °C for 35 min, was passed through the sealing pad of the flask for solid-phase microextraction (SPME) for 40 min. Three parallel experiments were conducted.

The GC conditions: DB-Wax column (30 m × 0.25 mm × 0.25 μm); the inlet temperature was 250 °C; high-purity He (purity ≥99.999%) carrier gas; the flow rate was 1000 mL/min; no shitter injection; initial temperature 45 °C for 2 min; 4 °C/min rose to 250 °C for 3 min.

MS conditions: Elector ionization (EI) mode of electron bombardment; ionization energy was 70 eV; ion source temperature was 230 °C; interface temperature was 280 °C; full-scan monitoring mode; mass scan range was 30–500 m/z.

Qualitative method [30]: We manually analysed the mass spectrum information of the sample, matched it with the Library Mainlib standard library and retained the positive and negative matching degrees of more than 800, or one volatile substance more than 900.

Quantitative method [30]: A semi-quantitative method was used in this study. We calculated the ratio of the peak area of the volatile to TMV to obtain its absolute concentration (assuming that the absolute correction factor of each volatile was 1.0) [31]. The calculation formula of the volatile matter concentration in the sample was as follows:$$Compound\;concentration/\left(ng/g\right)=\frac{V_{2}/V_{1}\times 0.2\times 0.92\times 1}{m}\times 1000$$
where *V_2_* is the peak area of volatile substance, *V_1_* is the peak area of the internal standard, 0.2 is the concentration of the internal standard (μL/mL), 0.92 is the density of the internal standard (mg/ μL), 1 is the volume of the internal standard (μL), and *m* is the mass of the sample 5 g.

### 2.4. Confirmation of Simulation System and Reaction Substrate

The SPME-GC-MS detection results of the reaction products of the three simulation systems were compared with the main aldehyde flavour substances in CWC in the literature [2] to confirm the experimental simulation system and the reaction substrate for the formation of phenylacetaldehyde.

### 2.5. Determination of the Optimum Proportion of Reaction Substrate

With reference to method 2.2, the molar ratios of glucose to phenylalanine in the experimental simulation system were made into 1:1, 1:2, 1:3, 1:4, 1:5, respectively, and steamed for 30 min. The reaction products of the simulation system were tested with reference to method 2.3, and the optimal reaction ratio was determined based on the yield of phenylacetaldehyde.

### 2.6. Determination of α-DCs

With reference to the literature [32], according to the quality of glucose, 0.7236 g OPD was added to the experimental simulation system and simulation system 2, respectively. The experiment was carried out with reference to method 2.2. OPD captured the α-DCs, the intermediate products of the simulation system, which should not be directly detected by GC-MS, to generate stable quinoxalines (Figure 1). Then, quinoxalines were detected according to the method in Section 2.3, and the quinoxalines were analysed according to the structure of quinoxalines in Figure 1. α-DCs were obtained.

## 3. Results and Analysis

### 3.1. Reaction Substrate for the Formation of Phenylacetaldehyde

The SPME-GC-MS results of the reaction products of the three simulated systems are shown in Table 1. It can be seen from Table 1 that the main aldehydes produced by simulation system 1 are consistent with the aldehydes produced by CWC cooking in the literature [2], and only simulation system 1 generates phenylacetaldehyde, the characteristic flavour substance of CWC. Therefore, simulation system 1 was selected as the experimental simulation system to study the formation mechanism of phenylacetaldehyde in the steaming process of CWC. It can also be found from Table 1 that the relative content of phenylacetaldehyde is the highest among the aldehydes produced in simulation system 1 and increases with the increase in steaming time (Figure 2). This indicates that phenylacetaldehyde is formed by a Maillard reaction of glucose and phenylalanine, which are the reaction substrates for the formation of phenylacetaldehyde. Phenylacetaldehyde was formed at 0 min in Figure 2, which indicates that the Maillard reaction between glucose and phenylalanine could take place when solid-phase microextraction is carried out at 80 °C.

### 3.2. The Optimal Molar Ratio of the Substrate to the Reaction

According to simulation system 1, the relative content of phenylacetaldehyde generated by the reaction of glucose and phenylalanine with different molar ratios of substances is shown in Figure 3. With the increase in the molar ratio of phenylalanine, the relative content of phenylacetaldehyde increased. When the molar ratio of glucose to phenylalanine was 1:4, the relative content of phenylacetaldehyde was close to the maximum, and when the molar ratio of the two substances was 1:5, the relative content of phenylacetaldehyde increased slowly. Therefore, the molar ratio of glucose to phenylalanine is 1:4, which is the best proportion of substances to generate phenylacetaldehyde. According to 2.2, the molar ratio of glucose to phenylalanine is 200.67:1, so the amount of glucose in simulation system 1 is greatly excessive. Determining the optimal ratio of the two substrates involves determining how much glucose is involved in the Maillard reaction with phenylalanine. According to the results, about 2.75 mg of glucose was involved in the Maillard reaction with phenylalanine in the simulated system 1, the glucose was greatly excessive, and the α-DCs mainly came from the decomposition of glucose. The optimal reaction ratio of glucose and phenylalanine can provide a reference for how many molecules of α-DCs are produced on average per glucose molecule.

### 3.3. The Main α-DCs Formed

As can be seen from Table 2, α-DCs in the experimental simulation system included MGO, BD, GO, PD and OP, and the sum of their relative contents increased with the increase in the steaming time, which was consistent with the trend of the relative content of phenylacetaldehyde increasing with the increase in the steaming time (Figure 2). When α-DCs were captured by OPD in the experimental simulation system, no phenylacetaldehyde was detected, indicating that the intermediate product of phenylacetaldehyde was α-DCs. Table 3 shows that the main α-DCs in simulation system 2 were MGO, BD, GO and PD. It can be seen that the two simulation systems generate relatively high contents of MGO and BD from 0 min, namely, 40 min extraction at 80 °C, indicating that MGO and BD mainly come from the caramelization reaction of glucose by comparing the reactants of the two simulation systems and the products in Table 2 and Table 3. The relative contents of MGO and BD in the experimental simulation system were slightly higher than that in simulation system 2, indicating that only a small amount of MGO and BD came from the Maillard reaction between glucose and phenylalanine. The relative content of GO ranked third among the intermediates, and the relative content of GO in the two simulation systems was similar, indicating that GO mainly comes from the caramelization reaction of glucose. GO is produced after 20 min of steaming, while MGO and BD are produced at 0 min of steaming, indicating that MGO and BD are more likely to be produced in a caramelization reaction. The relative content of PD in the simulation system was about 50% of that in the experimental simulation system, indicating that both the caramelization and Maillard reactions can generate PD, but the relative content of PD in the two simulation systems was small, and the contribution of PD to the generation of the characteristic flavour substance phenylacetaldehyde was small. OP only existed in the experimental simulation system and was derived from the Maillard reaction between glucose and phenylalanine, but its relative content was small, and it did not contribute much to the formation of the characteristic flavour substance phenylacetaldehyde.

Therefore, the main α-DCs in the experimental simulation system were MGO, BD and GO, which mainly came from the caramelization reaction of glucose. Their relative content was high, and they contributed the most to the formation of the characteristic flavour substance phenylacetaldehyde.

### 3.4. Analysis of Phenacetaldehyde Formation Mechanism in Maillard Reaction System

#### 3.4.1. The Formation Mechanism of Major α-DCs

According to the analysis results in Section 3.3, the formation mechanism of MGO, BD and GO can be analysed. The 3-DG generated by the enolization of glucose under acidic conditions and G generated by oxidation retains the complete hexose skeleton. These long-chain α-dicarbonyl substances are easily broken to form small-molecule α-DCs through enolization and oxidation reactions.

(1) MGO is mainly produced by the caramelization of glucose, and 3-DG is generated by the enolation of glucose through 1,2-enolation (Figure 4) [33]. The reverse aldol condensation of 3-DG at C3-C4 directly produces MGO [18,34] (Figure 5). Under the condition of excessive glucose in CWC, this is the main pathway.

(2) There are three GO formation mechanisms:

① G produced by glucose oxidation can form GO through C2-C3 cleavage or reverse aldol condensation [35,36,37] (Figure 6). Another product, 2,3,4-trihydroxybutyral, can undergo C2-C3 reverse aldol condensation and an oxidation reaction to produce GO (Figure 7).

② In Figure 5, the product of the reverse aldol condensation of 3-DG in C3-C4, 2,3-dihydroxypropanal can produce GO through the reverse aldol condensation and oxidation of C2-C3 (Figure 8).

③ GO can be generated through reverse aldol condensation in C3-C4 and oxidation reactions of glucose (Figure 7 and Figure 9). GO was produced only after 30 min of steaming, indicating that the glucose in CWC could not easily produce G, but was more likely to produce 3-DG, and the generated 3-DG mainly produced MGO.

(3) There are two mechanisms of BD formation: under the condition of excessive glucose, the main method of BD formation is the oxidation and elimination of 2,3,4-trihydroxybutyral produced by the reverse aldol condensation of glucose [13]. In addition, glucose can be polymerized into oligosaccharides, which produce Amadori products and then undergo a “peeling-off” mechanism to generate 1,4-dideoxy-2,3-hexadiketolose, which is then subjected to reverse aldol condensation to generate BD (Figure 10) [38,39].

#### 3.4.2. α-DCs React with Phenylalanine to Form Phenylacetaldehyde

In this study, the reaction mechanism of α-DCs with phenylalanine to produce phenylacetaldehyde is shown in Figure 11 [20]. The amino group of L-phenylalanine first underwent a nucleophilic addition reaction with the carbonyl group of α-DCs and removed one molecule of water to form imine I. After intramolecular rearrangement and decarboxylation, one molecule of CO_2_ was lost to form intermediate II. Then, α-aminoketones and the target product phenylacetaldehyde were obtained after hydrolysis.

In this study, the formation mechanism of phenylacetaldehyde in the Maillard reaction system is shown in Figure 12.

## 4. Conclusions

In the process of CWC steaming, glucose and phenylalanine in CWC undergo the Maillard reaction to produce the characteristic flavour substance phenylacetaldehyde. The formation mechanism was as follows: under the condition of steaming, glucose caramelized, mainly including the formation of 3-DG by enolization and the formation of G by oxidation. These long-chain α-DCs were converted into small-molecule α-DCs such as MGO, BD and GO by reverse aldol condensation and an oxidation reaction. Among them, MGO had the highest content and was the main product; then, α-DCs and phenylalanine underwent the Strecker degradation reaction to produce phenylacetaldehyde. The optimal molar ratio of glucose and phenylalanine to the Maillard reaction was 1:4. Glucose in CWC is greatly excessive, and phenylalanine can participate in the Maillard reaction completely.

## Figures and Tables

**Figure 1 foods-12-00498-f001:**
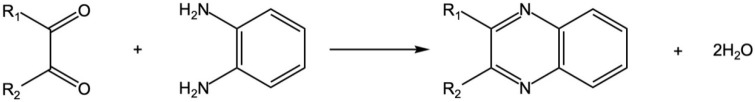
The reaction equation of OPD and α-DCs.

**Figure 2 foods-12-00498-f002:**
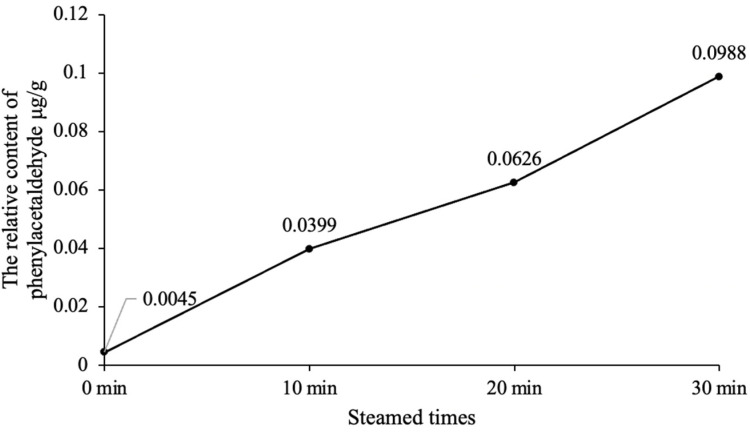
The relative contents of phenylacetaldehyde at different steaming times in simulated system 1.

**Figure 3 foods-12-00498-f003:**
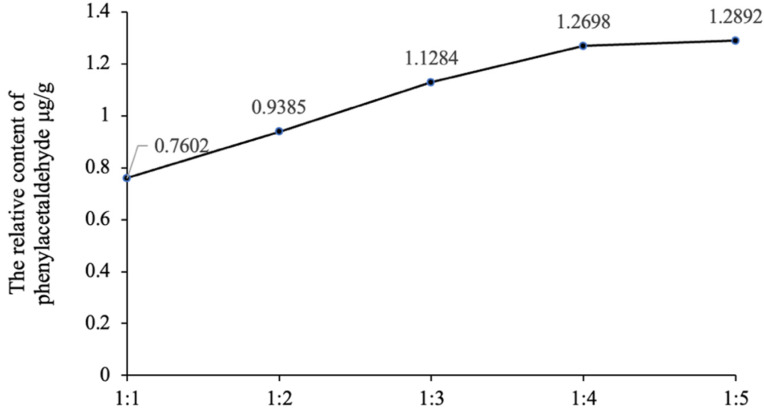
The relative content of phenylacetaldehyde produced by different reaction ratios of glucose and phenylalanine.

**Figure 4 foods-12-00498-f004:**
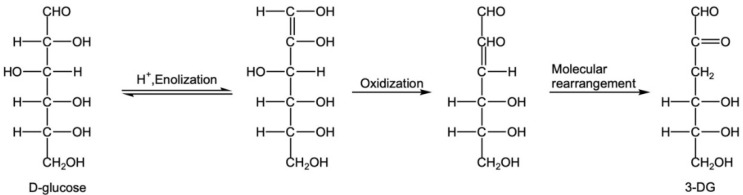
Enolization of glucose to produce 3-DG.

**Figure 5 foods-12-00498-f005:**
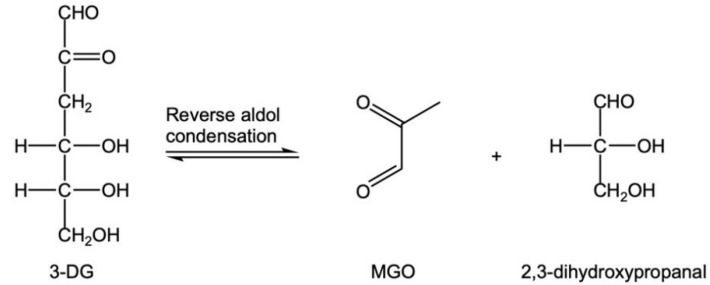
Reverse aldol condensation of 3-DG to produce MGO.

**Figure 6 foods-12-00498-f006:**
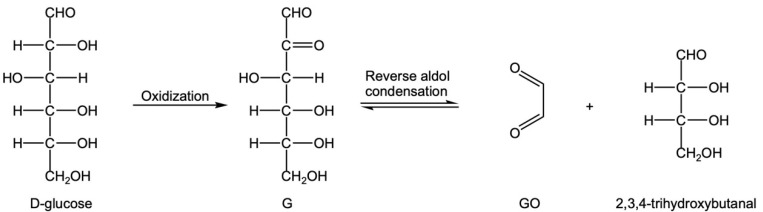
Reverse aldol condensation of G to produce GO.

**Figure 7 foods-12-00498-f007:**
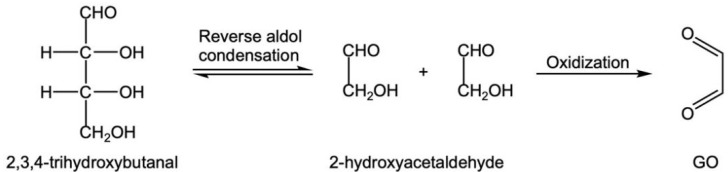
2,3, 4-trihydroxybutanal underwent reverse aldol condensation and oxidation reaction to produce GO.

**Figure 8 foods-12-00498-f008:**
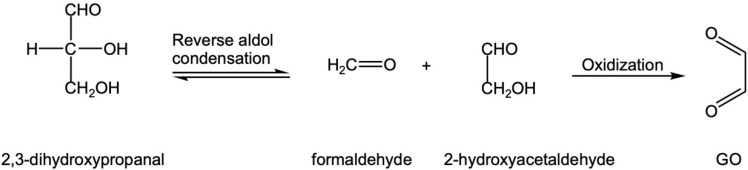
2,3-dihydroxypropanal underwent reverse aldol condensation and oxidation reaction to produce GO.

**Figure 9 foods-12-00498-f009:**
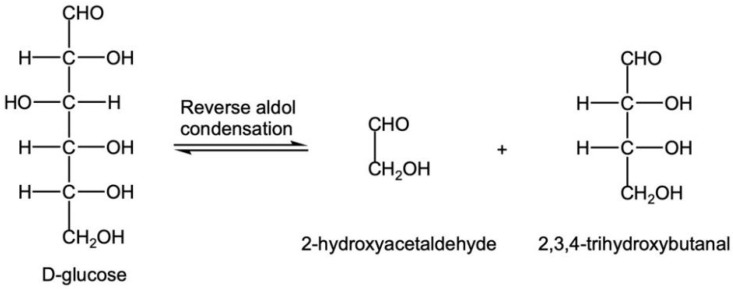
Reverse aldol condensation of glucose.

**Figure 10 foods-12-00498-f010:**
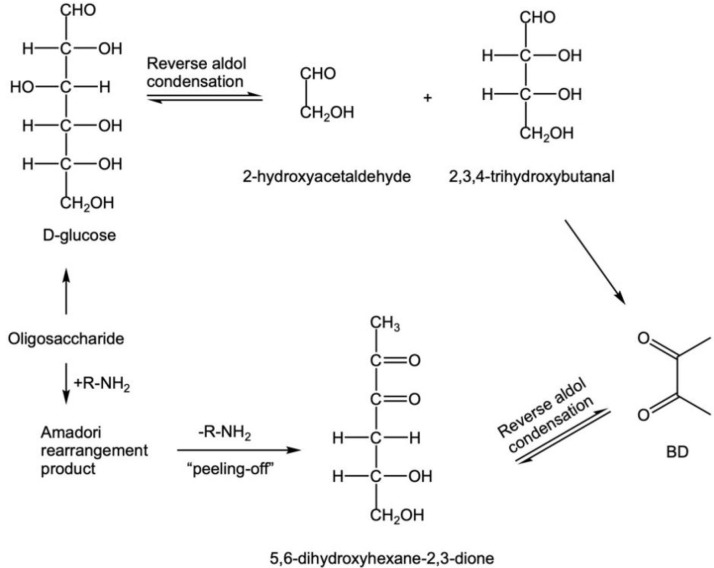
The glucose produces BD by reverse aldol condensation and “peeling-off” mechanism.

**Figure 11 foods-12-00498-f011:**
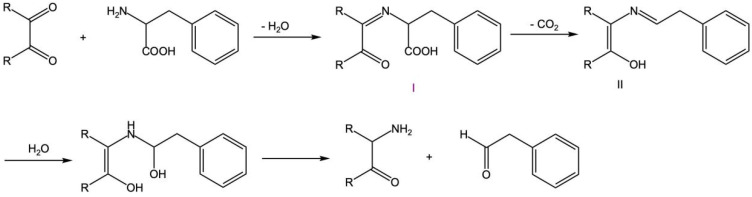
α-DCs react with L-phenylalanine to produce phenylacetaldehyde.

**Figure 12 foods-12-00498-f012:**
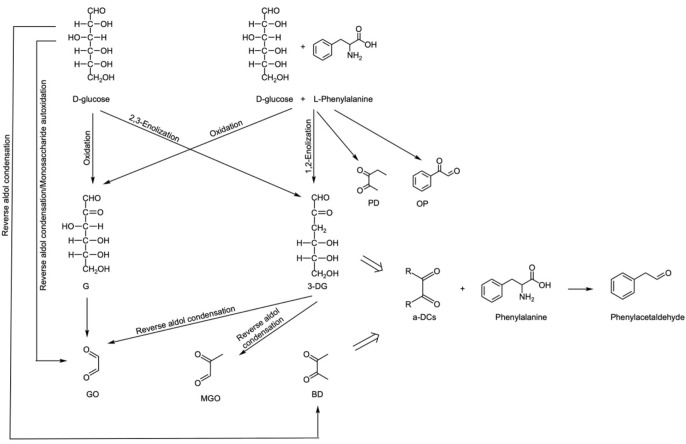
The mechanism of the reaction of glucose with phenylalanine to produce phenylacetaldehyde.

**Table 1 foods-12-00498-t001:** Main aldehydes and their contents in the reaction products of the simulated system.

Products	Relative Amount μg/g		
	Simulation System 1	Simulation System 2	Simulation System 3
Phenylacetaldehyde	0.0988	--	--
Benzaldehyde	0.0148	--	--
Capraldehyde	0.0097	--	--
Nonanal	0.0060	0.0109	0.0168
Dodecanal	0.0059	0.0047	0.0061
Undecanal	0.0018	0.0016	0.0047

Note: -- indicates that no relevant substance was detected.

**Table 2 foods-12-00498-t002:** Species and relative content of α-DCs in experimental simulation system (μg/g).

Steamed Time	Methylglyoxal(MGO) 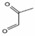	2,3-Butanedione(BD) 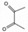	1-oxo-2-Phenylacetaldehyde(OP) 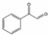	2,3-Pentanedione(PD) 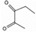	Glyoxal(GO) 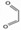	Total 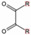
0 min	2.7990	0.1920	0.0094	--	--	3.0004
10 min	18.2800	2.3220	0.0214	--	--	20.6234
20 min	23.3401	4.2055	0.0289	2.7788	--	30.3533
30 min	27.5832	5.4023	0.0219	2.2209	4.9823	40.2106
Total	72.0023	12.1218	0.0816	4.9997	4.9823	94.1877

Note: -- indicates that no relevant substance was detected.

**Table 3 foods-12-00498-t003:** Species and relative content of α-DCs in simulation system 2 (μg/g).

Steamed Time	Methylglyoxal(MGO) 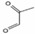	2,3-Butanedione(BD) 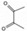	2,3-Pentanedione(PD) 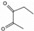	Glyoxal(GO) 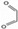	Total 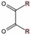
0 min	2.1846	0.2382	--	--	2.4228
10 min	16.1886	1.8177	--	--	18.0063
20 min	23.1943	2.8934	0.8822	3.5043	30.4742
30 min	23.7674	2.7870	1.0387	2.9894	30.5825
Total	65.3349	7.7363	1.9209	6.4937	81.4858

Note: -- indicates that no relevant substance was detected.

## Data Availability

The data will be available from the authors upon request.

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
