# Peer review of "Study on the Mechanism of Phenylacetaldehyde Formation in a Chinese Water Chestnut-Based Medium during the Steaming Process"

_foods, 2023, doi:10.3390/foods12030498_

Round 1

Reviewer 1 Report

Line 113-121 mention 3 simulation systems . However, the content of these 3 systems is the same. More detailed information should be given on this subject.

The title does not match the content. In the title, it is understood that production will be made from Chinese water chesnut (CWC), but there is a model system in the content.

On the other hand, as stated in the study, “there is information on aldeyhdes produced by CWC cooking in literatures (references such as 1 and 24). What is the difference between these studies and the current study? Plese explain.

The purpose of the study should be better explained.

Author Response

Dear Reviewers,

Manuscript ID: Foods-2093749

Title: Study on the Mechanism of Phenylacetaldehyde Formation in the Chinese Water Chestnut Based Mediums during Steaming Process

We are grateful for the comments and thoughtful suggestions addressed by reviewers. All those comments are valuable and very helpful for revising and improving our paper, as well as the important guiding significance to our research. We reviewed the comments carefully and made corrections accordingly. All the revised phrases and sections in the manuscript are marked in red color. We sincerely hope it will meet the requirements of the journal. The main corrections in the paper and the responds to the reviewers’ comments are listed as follows:

Point 1. Line 113-121 mention 3 simulation systems. However, the content of these 3 systems is the same. More detailed information should be given on this subject.

Response 1: Thanks for your comment, and the specific contents of these three simulation systems are as follows: System 1: 2.408/100 g glucose and 0.011/100 g phenylalanine were weighed and dissolved in 0.2 mol/L buffer solution of disodium hydrogen phosphate and sodium dihydrogen phosphate at pH 5.4, and then fixed volume to 100 mL. System 2: 2.408/100 g glucose was weighed and dissolved in the buffer solution above. System 3: 0.011/100 g phenylalanine was weighed and dissolved in the buffer so-lution above. The three simulation systems were compared to each other to determine the substrate of the target product phenylacetaldehyde. The results showed that glucose and phenylalanine could not form phenylacetaldehyde alone. Phenylacetaldehyde is produced by the Maillard reaction between the two.

Point 2. The title does not match the content. In the title, it is understood that production will be made from Chinese water chesnut (CWC), but there is a model system in the content.

Response 2: Thank you very much for your suggestion. The title has been changed to “Study on the Mechanism of Phenylacetaldehyde Formation in the Chinese Water Chestnut Based Mediums during Steaming Process”.

Point 3. On the other hand, as stated in the study, “there is information on aldeyhdes produced by CWC cooking in literatures (references such as 1 and 24). What is the difference between these studies and the current study? Please explain.

Response 3: Thanks for your comment, the reference 1 used e-nose technology to analyze the volatile flavor substances of Chinese water chestnut at different steaming and cooking time, and to evaluate the changes of the volatile flavor substances of Chinese water chestnut at different steaming and cooking time on the basis of sensory evaluation. And the reference 24 was used SPME-GC-MS technology to analyze the changes of volatile substances in Chinese water chestnut fresh and processed in different ways, and a method for determination of volatile substances in Chinese water chestnut was established. The present study is based on the results of the two references, we used the flavor substances produced by the steaming of Chinese water chestnut in reference 1 to determine its characteristic flavor substance phenylacetaldehyde by OAV value method, and used the detection method in reference 24 to detect the flavor substances in the three simulation systems to determine the reaction substrate of phenylacetaldehyde, and then carried out the next experiment to determine the formation mechanism of phenylacetaldehyde.

Point 4. The purpose of the study should be better explained.

Response 4: Thank you very much for your suggestion. We have supplemented the purpose of the study as “we simulated the contents of glucose and phenylalanine in CWC and its steaming processing conditions, captured the small molecules formed in the reaction process with OPD, detected α-DCs and the final products with SPME-GC-MS, determined the formation substrate and molecular transfer pathway of phenylacetaldehyde, to clarify the formation mechanism of the flavor substance phenylacetaldehyde. The results could provide a scientific reference for the study on the controlled release of the flavor substances and the evaluation and regulation of the flavor quality in the process of CWC steaming, and had important significance for improving the flavor quality of processed CWC products”.

Reviewer 2 Report

The recommendations and corrections were addressed on the manuscript file.

Author Response

Dear Reviewers,

Manuscript ID: Foods-2093749

Title: Study on the Mechanism of Phenylacetaldehyde Formation in the Chinese Water Chestnut Based Mediums during Steaming Process

We are grateful for the comments and thoughtful suggestions addressed by reviewers. All those comments are valuable and very helpful for revising and improving our paper, as well as the important guiding significance to our research. We reviewed the comments carefully and made corrections accordingly. All the revised phrases and sections in the manuscript are marked in red color. We sincerely hope it will meet the requirements of the journal. The main corrections in the paper and the responds to the reviewers’ comments are listed as follows:

Point 1. I am suggesting change in the title. For better following my suggestion like below: "Study on the Mechanism of Phenylacetaldehyde Formation in the Chinese Water Chestnut Based Mediums during Steaming Process".

Response 1: Thanks for your comment, and we have changed the title to “Study on the Mechanism of Phenylacetaldehyde Formation in the Chinese Water Chestnut Based Mediums during Steaming Process”.

Point 2. Please add reference/references for the statements.

Response 2: Thank you very much for your suggestion. We have added the relevant references.

Point 3. When preparing of CWC based medium, did the researchers followed the previous studies? If so, please add the reference/references.

Response 3: Thank you very much for your suggestion. According to the review of relevant literature (Zhao, T. P. et al), it was found that phenylalanine and glucose would undergo Maillard reaction to produce Phenylacetaldehyde at high temperature. Our research group used UPLC-MS/MS earlier and found that the contents of phenylalanine and glucose were significantly reduced under high temperature steaming and cooking, indicating that they reacted to produce other flavor substances in the steaming and cooking process. Subsequently, we determined the contents of phenylalanine and glucose in Chinese water chestnut by HPLC, and constructed three simulation systems to study the formation mechanism of Phenylacetaldehyde.

Point 4. What about temperature of the samples during treatment. this point is very critical for flavor chemistry. Temperature of the samples must be measured and shared for better evaluation of the collected results.

Did the researchers take into consideration of "come up time" to the target temperature?

Response 4: Thanks for your comment, according to the measurement, the steam environment in the steamer at this time is 100 degrees Celsius. We took the "come up time" into consideration during the test, so we added the same amount of water into the steamer each time in the test, and the limit power was 2200W. The boiling time of water was basically the same each time. After the water was boiling, the power was adjusted to 800W and the samples were immediately put in for steaming, so the boiling time of water was the same each time.

Point 5. Is the method mentioned below validated? Is the method mentioned below validated? Please add Recovery, LOD and LOQ values of the method.

Response 5: Thank you very much for your suggestion. According to the optimized method in the reference 26, we only used GC-MS to detect the flavor substances produced by the simulation systems, and did not detect the LOD and LOQ in this process. The GC-MS detection method in this study did not use working curve, so it is not suitable to calculate the recovery, and the relative content and yield of Phenylacetaldehyde were measured by internal standard method.

Point 6. I missed the aim behind determination of the optimum proportion of reaction substratethis processing. Please explain.

Response 6: Thanks for your comment, determining the optimal ratio of the two substrates is to determine how much glucose is involved in the Maillard reaction with phenylalanine. According to the results, the glucose is greatly excessive, and the α-DCs mainly come from the decomposition of glucose. The optimal reaction ratio of glucose and phenylalanine can provide a reference for how many molecules of α-DCs are produced on average per glucose molecule.

Round 2

Reviewer 1 Report

Necessary explanations and corrections were made by authors. It can be accepted for the publication.

Author Response

Dear professor,

Thanks for your review and revision suggestions.

Reviewer 2 Report

The authors improved the quality of the manuscript by following the the addressed recommendations. Therefore, there is no any comment.

Author Response

(The authors gave the same response as above.)
